# Observed feeding behaviours and effects on child weight and length at 12 months of age: Findings from the SPRING cluster-randomized controlled trial in rural India

**Pauline Boucheron**[1]*, **Sunil Bhopal**[1], **Deepali Verma**[2], **Reetabrata Roy**[1,2], **Divya Kumar**[2], **Gauri Divan**[2], **Betty Kirkwood**[1]

1 Maternal & Child Health Intervention Research Group, Department of Population Health, Faculty of Epidemiology & Population Health, London School of Hygiene & Tropical Medicine, London, United Kingdom, 2 Child Development Group, Sangath, New Delhi, India

* Pauline.boucheron1@alumni.lshtm.ac.uk

**Data Availability Statement:** All data files are available from the London School of Hygiene & Tropical Medicine (LSHTM) Data Compass

## Abstract

### Background

Child undernutrition results in poor growth in early childhood, undermines optimal development and increases the risk of mortality. Responsive feeding has been promoted as a key intervention for improving nutritional status, however measurement of this remains difficult and has rarely considered child behaviour. We therefore developed a new observed feeding tool to assess both child and caregiver behaviours, as well as their interaction during feeding, and investigate the effect of these on children anthropometric measures at 12-months of age in rural India.

### Methods

Our study was nested within the SPRING cluster-randomized controlled trial in Rewari, North India. Outcomes were children length-for-age (LAZ), weight-for-length (WLZ) and weight-for-age (WAZ) Z scores at 12 months of age, based on the WHO Child Growth standards. Trained non-specialists live-coded feeding episodes using the newly designed tool. Scores were then created using principal components analysis representing child behaviour, caregiver behaviour and caregiver-child interaction. Mixed effects linear regression was used to assess associations between feeding behaviours and anthropometric outcomes.

### Results

857 children had a meal observation and were included. Anthropometric status was poor (mean length-for-age -1.59 (SD = 1.11); mean weight-for-length -0.58 (0.95); mean weight-for-age -1.22 (1.04)). There were positive linear differences in weight-for-length per unit increase in caregiver responsive behaviours score (adjusted β-coeff = 0.006, 95%CI = (0.001, 0.011), p = 0.01), in length-for-age and weight-for-age per unit increase in child

Repository (https://doi.org/10.17037/DATA.00000947).

**Funding:** The work was funded by the Wellcome Trust (www.wellcome.ac.uk) through two awards: a Wellcome Trust Research Training Fellowship to SB (107818/Z/15/Z) & a Wellcome Trust programme grant for the SPRING Programme (0936115/Z/10/Z) for which BRK is the principal investigator. The funders had no role in study design, data collection and analysis, decision to publish, or preparation of the manuscript.

**Competing interests:** The authors have declared that no competing interests exist.

responsive behaviours score (respectively adjusted β-coeff = 0.004, 95%CI = (0.001, 0.007), p = 0.02, and adjusted β-coeff = 0.003, 95%CI = (0.00001, 0.006), p = 0.049), and in both weight-for-length and weight-for-age per unit increase in caregiver-child interaction score (respectively adjusted β-coeff = 0.007, 95%CI = (0.003, 0.012), p = 0.001, and adjusted β-coeff = 0.005, 95%CI = (0.001, 0.011), p = 0.01). No association was seen between child behaviours and weight-for-length, caregiver behaviours and length and caregiver-child interaction and length.

## Conclusions

We found that trained non-specialists could assess feeding episodes using a newly designed checklist. Further, child and caregiver behaviours were associated with weight and length at only 12 months of age, a reminder of the importance of interventions to improve responsive feeding quality as we strive towards achievement of the sustainable development goals.

## Introduction

Worldwide, about 150 million children under 5 years are stunted, 110 million underweight and 50 million wasted [1,2]. Undernutrition in early childhood is a major public health issue, especially in low and middle-income countries (LMICs), and leads to growth faltering which is related to impaired cognitive and socio-emotional development in children. Before the age of 2 years [3], a short length is a strong predictor of reduced schooling, poorer mental health in adolescence, shorter height and lower productivity in adulthood, as well as lower offspring birthweight [4–6]. This contributes to an intergenerational transmission of lost human capital and poverty [7,8]. Moreover, the poorer immunity seen in low weight children increases their risk of infectious disease-related mortality [9]. Improving early childhood growth is therefore crucial to reaching the Sustainable Development Goals 1 to 4, which aim to reduce poverty and undernutrition, to promote health and well-being at all ages, and provide inclusive access to education [10,11]. The period of greatest vulnerability to undernutrition is from around six months, when exclusive breastfeeding becomes insufficient to meet a child's nutritional requirements, until that point when the child can both self-feed and eat the same diet as the rest of the family [12]. This is usually by the age of 1 year old [13].

One solution to undernutrition may be responsive feeding. "Responsive feeding" is the result of applying principles of psychosocial care to the feeding situation [14,15], and is the name given to supportive carer behaviours during feeding, which are important to encouraging infants and young children to feed adequately. Specific responsive feeding behaviours recommended by the WHO include 1) feeding infants directly and assisting older children to self-feed, 2) being sensitive to child cues, 3) encouraging children to eat, 4) feeding slowly and patiently, 5) not practicing force-feeding, 6) trying other encouragement strategies when children refuse food, 7) minimizing distraction, and 8) interacting socially with children during meals [16]. These have been integrated into recent work on nurturing care in the recently launched *Nurturing Care for Early Childhood Development Framework* presented at the World Health Assembly, as a component of responsive caregiving [17].

However, assessing Infant & Young Child Feeding programmes with a focus on these aspects is difficult. This is because there is no method which allows non-specialists to perform

a holistic responsive feeding assessment; previous studies used a broad range of methods, from assessing one individual behaviour (e.g. hand-feeding [18]) or evaluating separately various categories of behaviours [19], to developing scales combining multiple behaviours (e.g. an active feeding scale including encouragement, threatening, serving or offering additional food, demonstrating how to eat more, and ordering the child to eat more [20]) or different components of feeding (e.g. a care index that includes type of food offered to the child, two responsive feeding behaviours, and use of preventive health care services [21]). However, most work focuses on caregivers over child behaviours.

We therefore created a new observational tool adapted for community-based interventions in low-resources settings that was suitable for 1) use with one-year-old children 2) administration in the home 3) assessment over one mealtime 4) administration by non-specialists and 5) live coding and scoring, to investigate the relationships between child & caregiver behaviours during feeding with length and weight at 12 months in Haryana state, India—the country where the prevalence of both stunting and wasting is highest. In this paper, we first present a method that uses a new scoring system to develop three indices for measurement of 1) child behaviours 2) caregiver behaviours and 3) caregiver-child interaction during feeding from a larger tool. We then quantify the association of each set of behaviours with 1) length-for-age, a marker of child long-term nutritional status that evolves gradually over time, 2) weight-for-length, an indicator of child current nutritional status prone to short-term variations, and 3) weight-for-age, a summary measure for both length-for-age and weight-for-length, with short-term variations reflecting changes in weight-for-length, and long-term variations reflecting changes in length-for-age [22].

## Methods

### Overview of SPRING trial study design

This analytical cross-sectional study was done within the SPRING cluster-randomised controlled trial in India. Details on SPRING are presented elsewhere [23] but in brief, SPRING in India was an innovative home visiting programme, delivered by community-based agents who used coaching techniques to support families to improve nutrition and responsive caregiving within households. The intervention was designed from the outset to be feasible, affordable and scalable through the national healthcare system. The aim was to improve growth and development through pregnancy and the first two years of life. SPRING was evaluated by cluster randomised controlled trial, with clusters designed to minimize the risk of contamination. There were 24 clusters representing catchment areas of functioning health sub-centres, the lowest level of the primary healthcare system. Clusters were allocated to intervention and control groups with a 1:1 ratio, using restricted randomisation. Both groups had access to routine maternal and child healthcare services. Primary outcomes for the trial were length-for-age Z-score, and the motor, cognitive and language scales of Bayley Scales of Infant Development III (BSID-III) [24], all measured at 18 months of age. This impact evaluation was complemented with an economic analysis and a process evaluation, which will provide a better understanding of the SPRING mechanisms of action and inform governments for scaling-up and incorporating the intervention into healthcare systems. The SPRING trial is registered with ClinialTrials. gov, number NCT02059863.

SPRING was implemented in 120 villages of three administrative areas of Rewari, a predominantly rural district of Haryana state, in North India, which represents a total population of around 200,000. In Rewari, demographic and health indicators are around average for Haryana state, with a female literacy rate of 67% for an overall literacy of 76%; a sex ratio amongst the lowest in India, with 879 females per 1000 males; and an infant mortality close to

the national average (41/1000 births) [25]. Rewari is covered by the Integrated Child Development Services, which provides complementary nutrition to all pregnant and lactating women and children [26]. Although Rewari is considered food secure, the prevalence of stunting in children under-five years old is extremely high, at 46% according to the SPRING baseline survey in 2014 (unpublished project data).

One sample size calculation was done for the SPRING-ELS substudy of which the observed feeding tool formed a part. The aim was to explore the effect of adversity on growth [25]. A minimum of 25 children per cluster was needed to give 90% power at the 5% level of significance to detect effect sizes between 0.4SD & 0.5SD, assuming an intra-cluster correlation of 0.05, using an established formula [27]. The work described in this paper used all available data, exceeding this calculated number.

## Data collection

A trial surveillance system was implemented whereby trained resident fieldworkers visited all households with women of reproductive age, married and not sterilised every 8 weeks to enrol pregnant women and newborns, and to follow up those already identified. Babies with major congenital malformations and maternal deaths in the neonatal period were excluded. Sociodemographic data were collected at enrolment by surveillance system fieldworkers using mobile phones.

A separate group of non-specialist assessors did assessments when enrolled children turned 12 months of age (within -7 to +21 days of this birthday). They had minimal experience of research, child assessment or use of observational tools. These assessments took around 2.5 hours and were spread over 2 days.

The assessors did anthropometrical measurements of infants at 12 months. They measured weight with a precision of 0.01Kg using SECA-384 electronic scales calibrated each week. Ideally, children were weighed with their clothes removed. In cases where this was not possible, children were weighed fully-clothed, then their clothes were removed and weighed. The child's weight was then calculated by subtracting the weight of the child's clothes to the weight of the fully-clothed child. Length was measured by two assessors with a precision of 0.1cm using the SECA-417 infantometer [25,28].

Assessors aimed to observe the caregiver and child during a meal if one was planned during the 3–5 hour period in which they were in the home doing other assessments, using our new Observed Feeding Tool. S1 Appendix presents the 34 items that this tool contains and details how each item was scored by observers, while S2 Appendix present standard operating procedure for the meal observation. Two items were scored before the meal, 11 during and 21 after the meal; it covers four elements of complementary feeding: 1) hygiene practices, 2) food quantity, 3) feeding behaviours and 4) food diversity. Assessor-expert reliability tests done using videos showed an overall reliability of 90% for all items with more than 80% agreement for each assessor. To limit the risk that caregivers and children change their behaviours because they knew they were observed, assessors had received instructions to sit in an unobtrusive position and to not intervene during the meal observation.

During the meal itself, the assessor ticked a box each time a mouthful of food entered the child's mouth, up to a maximum 30 times. Simultaneously, they observed 11 specific behaviours relating to caregiver encouragement, caregiver responsiveness, harsh behaviours and the child's response to food. They noted by ticking a box each time a specific behaviour occurred, up to a maximum of three times. After the meal, assessors estimated the volume of food consumed during the meal using a *katori*–a widely used small stainless steel bowl with a volume of 160mL, assessed whether the caregiver or child had ended the meal and recorded the caregiver

who was mainly in-charge of feeding the child. They asked this person questions related to reasons to start feeding, and whether the meal observed was typical, in terms of feeder, place of feeding and type of food. Finally, assessors evaluated social interaction during the meal (talking & singing), interruptions, whether the child had their own plate or bowl, persons eating with the child, feeding location and types of foods offered during the meal. Caregiver's behaviours, child's behaviours and caregiver-child interaction observed during the meal are the focus of this paper.

### Data analysis

**Outcomes.** Child length and weight were converted to Z-scores using the zscore06 package for Stata15 [29] based on the 2006 WHO Child Growth Standards [28]. Therefore, length-for-age Z-score (LAZ), weight-for-length Z-score (WLZ) and weight-for-age Z-score (WAZ) at 12 months were the three outcomes for this study, expressed as continuous variables. Z-scores represented the number of standard deviations from the mean when compared with WHO Child Growth Standards, which represent the gold standard to describe normal growth in healthy breastfed children irrespective of country, ethnicity, socioeconomic status and type of feeding [28]. We excluded from the analysis mother-child pairs whose children had missing or implausible values of anthropometry identified using standard rules used by the WHO [22].

**Exposures.** We created three feeding behaviour indices measuring a) child behaviours b) caregiver behaviours and c) caregiver-child interaction during feeding using data from the behavioural section of the observed feeding tool. Table 1 presents child and caregiver behaviours assessed by the observer during feeding that were included respectively in Index A and B. Index A (child behaviours) assessed child's interest in food, child's social interaction with the caregiver, whether the child had ended the meal and child's expression of hunger cues after the end of the meal. Index B (caregiver behaviours) assessed caregiver's behaviours towards self-feeding, caregiver's encouragements to promote eating, caregiver's reactions to child's cues or disinterest in food, caregiver's social interaction and attention to the child during the meal, caregiver's behaviours that distracts the child during the meal, harshness and whether the caregiver had ended the meal. All behaviours were included in Index C. The observer assessed separately behaviours within a category (e.g. 'promoting' and 'discouraging' self-feeding behaviours) using the observed feeding tool. Each behaviour assessed was converted into a binary variable. We then ran an unrotated principal component analysis with a single component using a correlation matrix because raw data was not standardized. We extracted the first principal component of each index, with the aim of capturing the linear combination of feeding behaviours within each index which creates the maximum variance of the data. This is a similar method to that commonly employed to calculate socioeconomic status indices [30]. The child feeding behaviour scores were reversed because positive scores here indicated poorly quality feeding behaviours. To enhance interpretability, the raw PCA score obtained for each index was standardized on a scale from 0 (the lowest PCA score) to 100 (the best PCA score).

**Descriptive statistics and handling of missing data.** We calculated descriptive statistics for all outcomes, exposures and potential confounders. We performed analysis in complete case analysis and compared baseline characteristics in children observed versus not observed during a meal, in order to assess the potential for selection bias.

**Modelling the association between feeding behaviours and anthropometric outcomes.** We used a causal backward modelling approach [31] to study the independent associations between the three feeding indices and the three anthropometry outcomes. We performed mixed-effects linear regression, accounting for clustered-design as a random effect and trial arm allocation as a fixed effect to calculate the adjusted mean growth value at each score of

Table 1. Items* of the Observed feeding tool included in feeding behaviours indices**,***.

| a) Index A: Child behaviours | b) Index B: Caregiver behaviours |
|---|---|
| **Child** | **Caregiver** |
| **1. Interest in food** | **1. Self-feeding** |
| 1. Tries to get food (e.g. by asking, pointing to food, reaching for food, touching food or opening mouth) 2. - Shows disinterest in having food (e.g. says no, sticks out tongue, closes mouth, turns or moves away) | 1. Encourages or helps self-feeding (e.g. by giving food to the child to eat themselves or clap hands) 2. Discourages or stops the child from self-feeding (e.g. by saying 'no' or taking food away from the child when they try to pick it up) |
| **2. Social interaction** | **2. Encouragement** |
| Interacts with caregiver during feeding (e.g. by laughing, talking about things apart from food, singing songs, touching caregiver, smiling, looking at caregiver) | **Verbal encouragement**: encourages the child to eat but not in response to the child, by saying things like 'eat, eat', 'chappati is nice', or 'you are so good' |
| | **Encouragement by playing**: encourages the child to eat by imitating feeding or playing positive food games |
| **3. Child ends meal (determined by looking at the last two mouthfuls of food)** | **3. Reacting to the child** |
| Child refused the last 2 mouthfuls or was self-fed and stopped independently | 1. Responds positively to child cues (e.g. child indicates food is too hot, and caregiver makes it cooler; or child wants more food and caregiver gives food) 2. - When child is bored, says 'no' or tries to stop feeding, caregiver tries to find positive strategies to keep child's interest in food (e.g. by offering another type of food or diverting child briefly) |
| **4. Meal ended prematurely** | **4. Harshness** |
| 1. Child showed signs of hunger after the meal has ended 2. Child consumed ≤4 mouthfuls | Force feeds, holds child's head still to give food, shakes child, threatens child, uses an angry tone of voice, shouts or berates child |
| | **5. Social interaction** |
| | Interacts with child during feeding (e.g. by laughing, talking about things apart from food, singing songs, touching child, smiling, looking at child) |
| | **6. Distraction** |
| | Encourages attention away from feeding (e.g. by stopping feeding or leaving the place during the meal) |
| | **7. Attention** |
| | Gives the child full attention during feeding |
| | **8. Caregiver ends meal (determined by looking at the last two mouthfuls of food)** |
| | Food was finished or child refused once, and caregiver ended the meal with no additional encouragement |

*Each item included in the indices were made binary, scored 1 if the behaviour was observed and 0 if it was not observed.

**Each index was scored on a scale from 0 (lowest score corresponding to the least responsive feeding behaviours) to 100 (highest score corresponding to the highest responsiveness during feeding).

***All child and caregiver items were included in the Caregiver-child interaction index.

behaviour indices. This allowed us to examine the change in these outcomes as children were exposed to incrementally greater scores of responsiveness. All models were adjusted for the following potential confounders: sociodemographic characteristics, maternal psychological risk factors, hygiene practices and feeding environment. We did not consider food quantity or food diversity as confounders because these were likely to be on the causal pathway [32,33].

We included child age and sex as forced variables because there was no risk of overfitting [34]. After running final models, we checked for multicollinearity and departure from linearity using variance inflation factor criteria and diagnostic plots. Lastly, we assessed linear interactions by *a priori* identified potential effect modifiers (child sex, maternal education and socioeconomic status) using likelihood ratio tests. All analyses were performed using Stata v15 (StataCorp LLC: College Station, TX, USA).

### Ethics statement

SPRING received ethical approval from the London School of Hygiene and Tropical Medicine (LSHTM) research ethics committee (SPRING: 23 June 2011, approval number 5983; SPRING-ELS substudy 19 May 2015, approval number 9886). Specific approval for this analysis was obtained from the London School of Hygiene and Tropical Medicine (LSHTM) MSc research ethics committee (8 May 2018, approval number 15508). SPRING also had approval from the Sangath Institutional Review board (IRB) (SPRING: 19 February 2014; SPRING-ELS substudy 27 May 2015) and from the Indian Council of Medical Research's Health Ministry Screening Committee (HMSC) (SPRING: 24 November 2014; SPRING-ELS substudy: 6 October 2015). We obtained informed written consent from mothers at enrolment into the trial surveillance system and before a child's first birthday.

This document complies with the STROBE guidelines.

## Results

### Sample description

The flowchart in Fig 1 shows that among 1,726 mother-child dyads eligible for SPRING outcome assessment, 874 had a meal observation. 422 mother-child dyads were lost to follow-up prior to this mainly because families were not available for assessment (12.0%), refused consent (5.9%), had moved away (4.2%) or because the mother or child had died (2.3%). Main reasons for not being observed during a meal included complementary feeding not being introduced yet (18.1%) (i.e child exclusively breastfed), no occurrence of a mealtime while assessors were in the household (4.6%), child sickness (1.5%), consent refusal (0.4%) and interruption of meal observation by the family (0.3%). Of the 874 mother-child pairs who had meals observed, 857 were included in this analysis; 16 mother-child pairs were not included because all required data were not available, and one child was excluded because of implausible anthropometric measures.

Table 2 presents sociodemographic characteristics of the 857 mother-child pairs observed during feeding. Overall, 442 children (51.6%) were males, and 20 (2.3%) were twins or triplets. Most children (n = 838, 97.8%) had been delivered in facilities, of mothers with a mean age at delivery of 22.3 (SD = 3.6) years. The majority of mothers had an education level of 10th to 12th grade (n = 335, 39.1%). Most meals observed were typical in terms of food provided (92.4%), feeder (81.9%) and place of feeding (76.2%). Children ate a median of 13 mouthfuls (IQR = 9;19) of food and most of them ate less than a 1/4 of a standard katori (n = 431/857, 50.3%), representing a volume of food of about 40mL. There was no evidence of selection bias in children in our study sample compared to those not observed, with regards to trial arm, maternal education, socioeconomic status, sex, maternal age at delivery, as well as delivery place. The proportion of twins/triplets differed, with a small p-value (p = 0.01); however, prevalence of twins/triplets was very low in both samples.

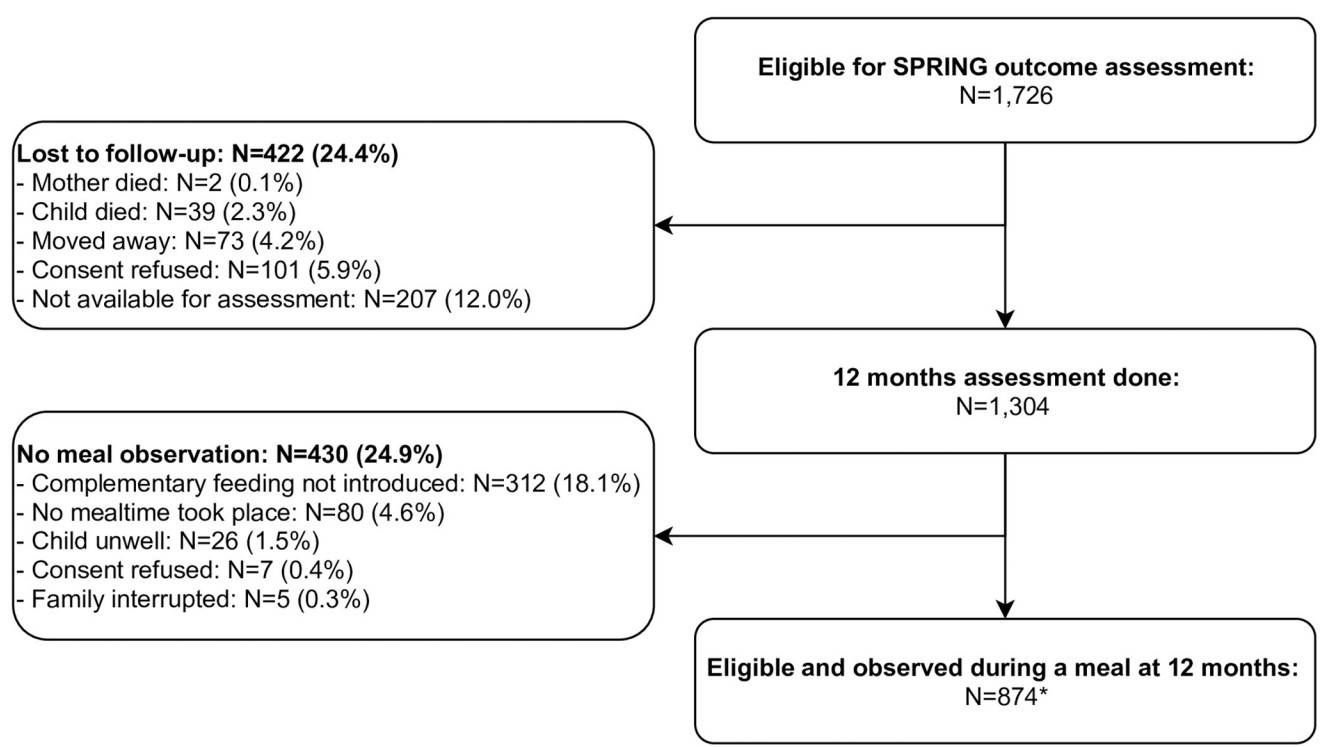

**Fig 1. Flowchart.** *17 mother-child pairs were excluded because of an implausible anthropometry (N = 1) or missing data (N = 16).

## Child anthropometry at the 12-month assessment

Fig 2 and Table 3 show that compared with an ideal mean of 0, the mean length-for-age was exceptionally low at -1.59 Z-score (SD = 1.11). This was similar for weight-for-length (-0.58 Z-score, SD = 0.95) and weight-for-age (-1.22 Z-score, SD = 1.04) showing that all three anthropometry measures were below WHO standards at age 12 months. Overall, 40% of children had signs of moderate to severe undernutrition, meaning that they had at least one anthropometric outcome with a Z-score ≤-2, with 35% being stunted (low length-for-age z-score), 23% underweight (low weight-for-age z-score) and 7% wasted (low weight-for-length z-score).

## Description of feeding behaviours

Table 4 and Fig 3 present characteristics of feeding behaviours. Of the three indices, the scores were higher in Index C (caregiver-child interaction) compared with Index B (caregiver behaviours) and Index A (child behaviours) (median scores of 73/100, 64/100 and 25/100 respectively). The principal component of Index A had an eigenvalue of 1.45 and explained 24.1% of the variability observed in the data. Similarly, these were respectively 1.66 and 15.1% for Index B, and 2.29 and 14.3% for Index C. The number of components with an eigenvalue > 1 were 2/6 for index A, 5/11 for Index B and 6/16 for Index C. The majority of caregivers verbally encouraged the child to eat (95%), interacted socially (93%) and gave full attention to the child (96%) during feeding. Uncommon behaviours observed in caregivers included harshness and distraction (each 2%), encouragement as well as discouragement of self-feeding (respectively 36% and 16%), being responsive to child cues (25%) and using positive strategies to overcome child refusal to eat (18%). Concerning children, most of them interacted socially during the meal (89%) and expressed more disinterest (75%) than interest in food (57%) while eating.

**Table 2. Comparison of children observed for a meal with those with no meal observation enrolled in SPRING.**

| Indicator | | Meal observed (O) | Meal not observed (N) | O-N Difference * (95% CI) | p-value |
|---|---|---|---|---|---|
| Children in sample | | 857 | 852 | | |
| Trial arm %(n) | Trial arm A | 48.3 (414) | 51.2(436) | -2.87(-11.26,5.53) | 0.49 |
| | Trial arm B | 51.7 (443) | 48.8(416) | 2.87(-5.53,11.26) | |
| Maternal education %(n) | ≤ 4 years | 11.4 (98) | 12.0 (102) | -0.54(-3.80,2.73) | 0.89 |
| | 5th to 9th grade | 24.7(212) | 24.1(205) | 0.68(-3.95,5.31) | |
| | 10th to 12th grade | 39.1 (335) | 38.0 (324) | 1.06(-3.67,5.79) | |
| | Higher education | 24.7 (212) | 25.9 (221) | -1.20(-5.25,2.84) | |
| Socioeconomic quintile %(n) | Q1 (lowest) | 21.2 (182) | 23.9 (204) | -2.71 (-7.19,1.78) | 0.60 |
| | Q2 | 18.1 (155) | 18.1 (154) | 0.01 (-3.41,3.43) | |
| | Q3 | 19.3 (165) | 19.8 (169) | -0.58 (-4.01,2.84) | |
| | Q4 | 21.2 (182) | 18.8 (160) | 2.46 (-2.28,7.19) | |
| | Q5 (highest) | 20.2 (173) | 19.4 (165) | 0.82 (-4.08,5.72) | |
| Mean SES score (SD) | | -0.01 (2.4) | -0.01 (2.6) | -0.01 (-0.34,0.32) | 0.97 |
| Male %(n) | | 51.6 (442) | 56.5 (481) | -4.88 (-10.52,0.76) | 0.09 |
| Twins/Triplets %(n) | | 2.3 (20) | 0.59 (5) | 0.88 (0.14,1.61) | 0.01 |
| Delivered in facility %(n) | | 97.8 (838) | 97.7 (832) | -0.13 (-1.49,1.23) | 0.85 |
| Mean age of mother at delivery (SD) | | 22.3 (3.6) | 22.4 (3.8) | -0.03 (-0.40,0.34) | 0.87 |
| Median number of mouthfuls of food eaten by children (IQR) | | 13 (9;19) | - | - | N/A |
| Number of Katoris of food eaten by children %(n) | < 1/4 | 50.3 (431) | - | - | N/A |
| | 1/4 | 31.5 (27) | - | - | |
| | 1/2 | 12.8 (110) | - | - | |
| | 3/4 | 1.2 (10) | - | - | |
| | ≥ 3/4 | 4.2 (36) | - | - | |

* Adjusted for clustering.

Most meals were child ended (55%) and 14% ended prematurely. Of note, SPRING trial arm allocation did not have a meaningful impact on feeding behaviours, with very small point estimates, wide confidence intervals and large p-values for each of the feeding scales (data not shown).

## Associations between observed feeding indices and anthropometry

Table 5 and Fig 4A, 4B & 4C present results of the univariate and multivariate linear regressions modelling the associations between the three feeding behaviours indices and the three anthropometric outcomes. All three indices were associated with anthropometric outcomes. After adjusting for confounding, associations were somewhat attenuated. Associations were strongest for weight-for-length and weight-for-age Z-scores with Index C (caregiver-child interaction) followed by Index B (caregiver behaviours) and weakest for length-for-age, which was only associated with Index A (child behaviours).

**Index A (child behaviours).** After adjustment, each point increase in Index A score was associated with a positive linear difference in length-for-age Z-score of 0.004 (95%CI 0.001, 0.007) (p = 0.02) (Table 5). This was 0.003 Z-score (95%CI 0.00001, 0.006) for weight-for-age (p = 0.049). For each 10 points increase in Index A score, this represented an average of 0.10cm in length gain (Fig 4A) and a mean weight-for-age gain of 30 g at 12 months (Fig 4C). There was no evidence for weight-for-length (p = 0.24).

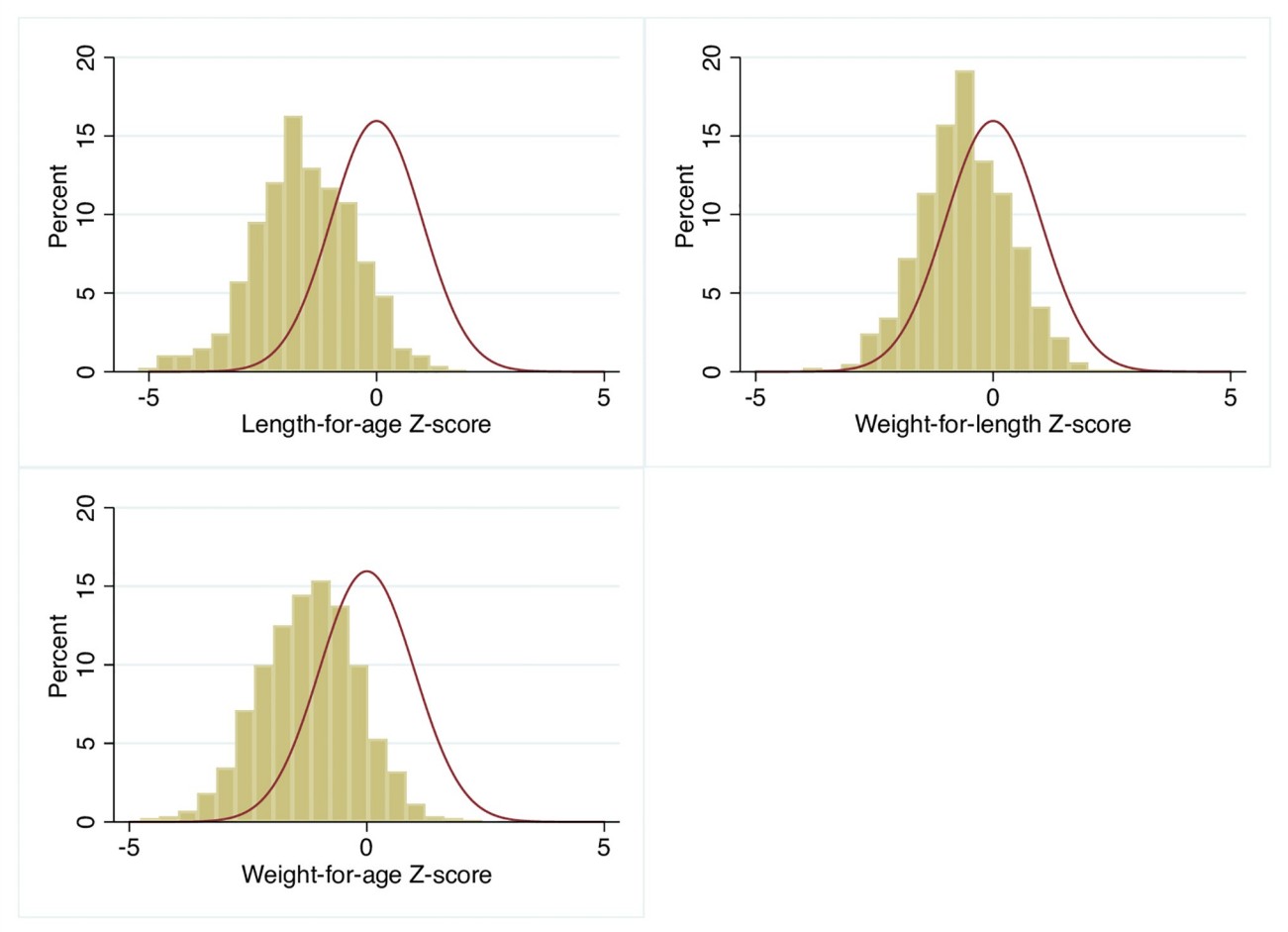

**Fig 2. Distribution of children length-for-age, weight-for-length and weight-for-age Z-scores at 12 months*** **as compared with the WHO Child Growth Standards****.** * Bars represent the distribution of children anthropometric outcomes at 12 months in this study. ** The red line represents the distribution of children anthropometric outcomes at 12 months in the WHO Child Growth Standards.

**Index B (caregiver behaviours).** After adjusting for confounding, there was some evidence of a positive linear difference in weight-for-length Z-score of 0.006 (95%CI 0.001, 0.011) per each unit increase in Index B score (p = 0.01) (Table 5). Considering children of average length at 12 months according to WHO child Growth standards, this represented a mean weight-for-length gain of about 48 g for each 10 points increase in Index B score (Fig 4B). There was very weak evidence of an association between Index B score and weight-for-age (p = 0.10) and no evidence for length-for-age (p = 0.94).

**Index C (caregiver-child interaction).** After adjusting for confounders, each point increase in Index C score was associated with a positive linear difference in weight-for-length

**Table 3. Description of child anthropometry at 12 months (N = 857).**

| Children anthropometric outcomes | Mean (SD) | N (%) Z-score $\leq$ -2 |
|---|---|---|
| Length-for-age z-score | -1.59 (1.11) | 300 (35.0) |
| Weight-for-length z-score | -0.58 (0.95) | 58 (6.8) |
| Weight-for-age z-score | -1.22 (1.04) | 197 (23.0) |

**Table 4. Description of child and caregiver behaviours observed during feeding (n = 857).**

| Behaviours | N | % |
|---|---|---|
| Child interacted socially with the caregiver | 779 | 89.3 |
| Child Showed disinterest in food | 656 | 75.2 |
| Child Showed interest in food | 494 | 56.7 |
| Child Ended meal | 482 | 55.3 |
| Child showed signs of premature end of the meal | 120 | 13.8 |
| Caregiver gave full attention to the child | 821 | 95.8 |
| Caregiver verbally encouraged child to eat | 812 | 94.7 |
| Caregiver interacted socially with the child | 793 | 92.5 |
| Caregiver encouraged child to self-feed | 310 | 36.2 |
| Caregiver responded positively to child cues | 211 | 24.6 |
| Caregiver found positive strategies to overcome child refusal | 155 | 18.1 |
| Caregiver encouraged child to eat by playing | 113 | 13.2 |
| Caregiver ended meal | 387 | 45.2 |
| Caregiver discouraged child to self-feed | 134 | 15.6 |
| Caregiver distracted child | 19 | 2.2 |
| Caregiver was harsh towards child | 13 | 1.5 |

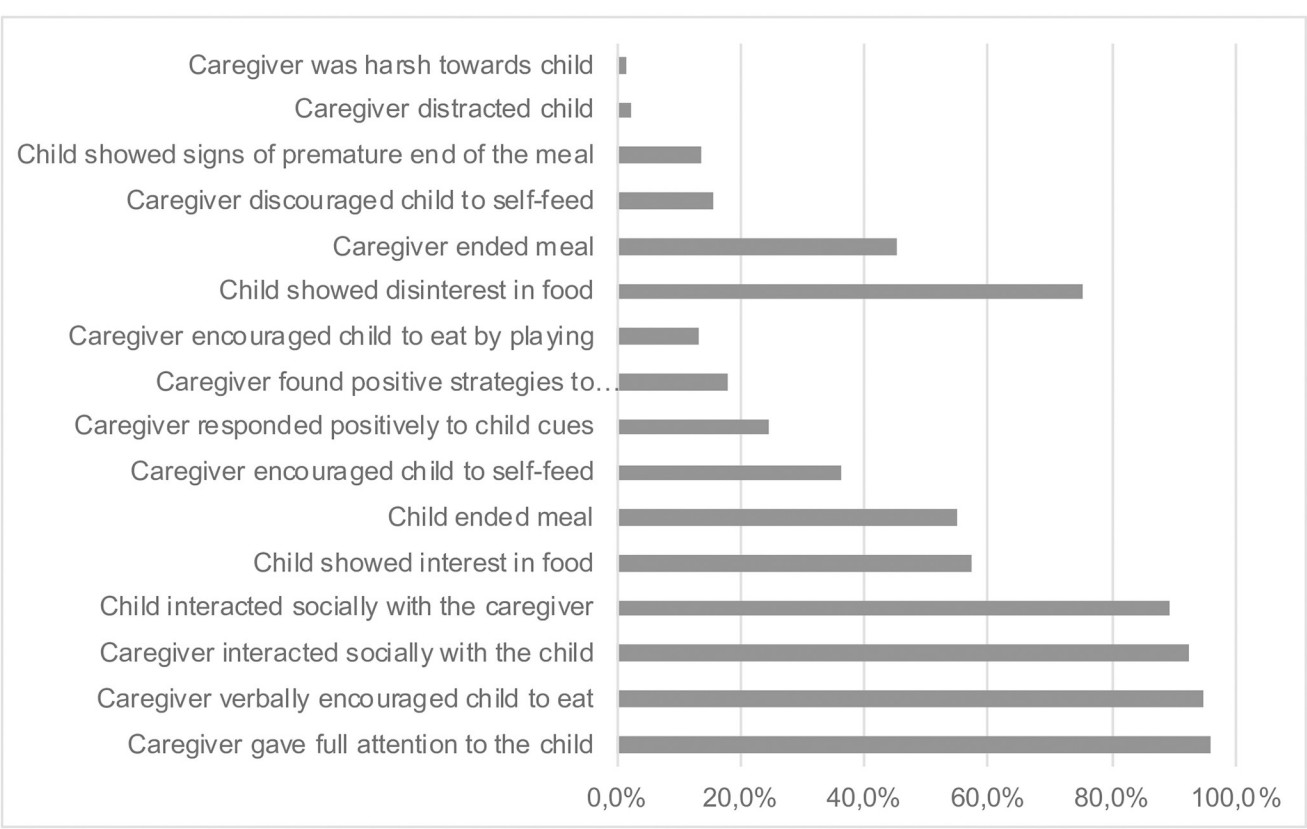

**Fig 3. Characteristics of behaviours observed during feeding (N = 857).**

**Table 5. Impact of feeding behaviours on children anthropometric status at 12 months.**

| Feeding behaviour indices | | Anthropometry | | |
| --- | --- | --- | --- | --- |
| | | Length-for-age Z-score | Weight-for-length Z-score | Weight-for-age Z-score |
| Index A (child behaviours) | Mean anthropometric outcome when child responsive feeding behaviour score is 0 (95%CI) | -1.76 (-1.92,-1.59) | -0.68 (-0.80,-0.56) | -1.38 (-1.52,-1.24) |
| | Crude difference in anthropometric outcome per unit increase in child responsive feeding behaviour score (β-coeff 95% CI)* | 0.005 (0.001,0.009) | 0.003 (0.00004,0.006) | 0.005 (0.001,0.008) |
| | p-value** for crude β-coeff | 0.01 | 0.053 | 0.01 |
| | Adjusted difference in anthropometric outcome per unit increase in child responsive feeding behaviour score (β-coeff 95% CI)*** | 0.004 (0.001,0.007) | 0.002 (-0.001,0.005) | 0.003 (0.00001,0.006) |
| | p-value** for adjusted β-coeff | 0.02 | 0.24 | 0.049 |
| Index B (caregiver behaviours) | Mean anthropometric outcome when caregiver responsive feeding behaviour score is 0 (95%CI) | -1.60 (-1.98,-1.23) | -0.88 (-1.19,-0.57) | -1.45 (-1.79,-1.11) |
| | Crude difference in anthropometric outcome per unit increase in caregiver responsive feeding behaviour score (β-coeff 95% CI)* | 0.0002 (-0.005,0.006) | 0.005 (0.0001;0.009) | 0.003 (-0.002,0.009) |
| | p-value** for crude β-coeff | 0.94 | 0.054 | 0.18 |
| | Adjusted difference in anthropometric outcome per unit increase in caregiver responsive feeding behaviour score (β-coeff 95% CI)*** | 0.0002 (-0.006,0.006) | 0.006 (0.001,0.011) | 0.004 (-0.001,0.009) |
| | p-value** for adjusted β-coeff | 0.94 | 0.01 | 0.10 |
| Index C (caregiver-child interaction) | Mean anthropometric outcome when overall responsive feeding behaviour score is t 0 (95%CI) | -1.83 (-2.23,-1.44) | -1.16 (-1.48,-0.84) | -1.76 (-2.11,-1.40) |
| | Crude difference in anthropometric outcome per unit increase in overall responsive feeding behaviour score (β-coeff 95% CI)* | 0.003 (-0.002,0.009) | 0.008 (0.004,0.013) | 0.008 (0.003,0.012) |
| | p-value** for crude β-coeff | 0.21 | 0.0003 | 0.003 |
| | Adjusted difference in anthropometric outcome per unit increase in overall responsive feeding behaviour score (β-coeff 95% CI)*** | 0.002 (-0.004,0.007) | 0.007 (0.003,0.012) | 0.006 (0.001,0.011) |
| | p-value** for adjusted β-coeff | 0.56 | 0.001 | 0.01 |

*Crude β-coeff (95%CI) obtained using linear regressions models with clusters as random effects

** Wald tests

***Adjusted β-coeff (95%CI) obtained using mixed linear regressions models with clusters as random effects and trial arm as fixed effects, adjusted for child age and sex, maternal education, socioeconomic quintile, maternal age at delivery, place of delivery, twins/triplets, caregiver's and child's handwashing before feeding, person who fed the child, reason to start feeding, place of feeding, eating with siblings, child having his own plate, quality of the mother-child dyad relationship (MORS-BF), maternal DUSOCS and PHQ9 scores

Z-score of 0.007 (95%CI 0.003, 0.012) (p = 0.001) (Table 5). This was 0.006 Z-score (95%CI 0.001, 0.011) for weight-for-age (p = 0.01). For each 10 points increase in Index C score, this represented a mean weight-for-age gain of 60 g at 12 months (Fig 4C) and of 56 g for weight-for-length in children of an average length according to WHO standards (Fig 4B). There was no evidence of an association between Index C score and length-for-age (p = 0.56).

## Discussion

We present data from the SPRING trial, where caregiver and child behaviours were observed during a meal. We found that the Observed Feeding Tool was suitable for assessment of feeding episodes by trained non-specialists. The key finding was that, at 12 months of age, positive child feeding behaviour was associated with increased length, and that positive caregiver behaviours and caregiver-child interaction were positively associated with child weight.

The Observed Feeding Tool was used by trained non-specialists to assess feeding episodes in a population of children aged 12 months with their mother in rural India. Assessor-expert reliability and agreement for each assessor were high suggesting that this tool may be suitable

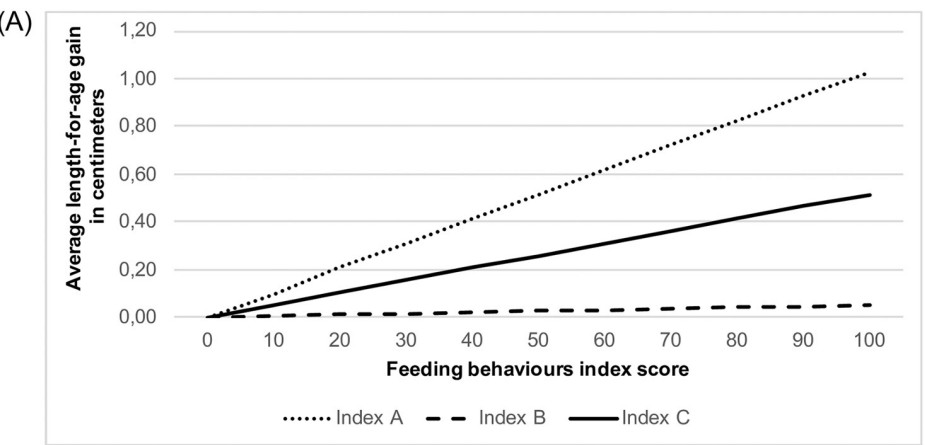

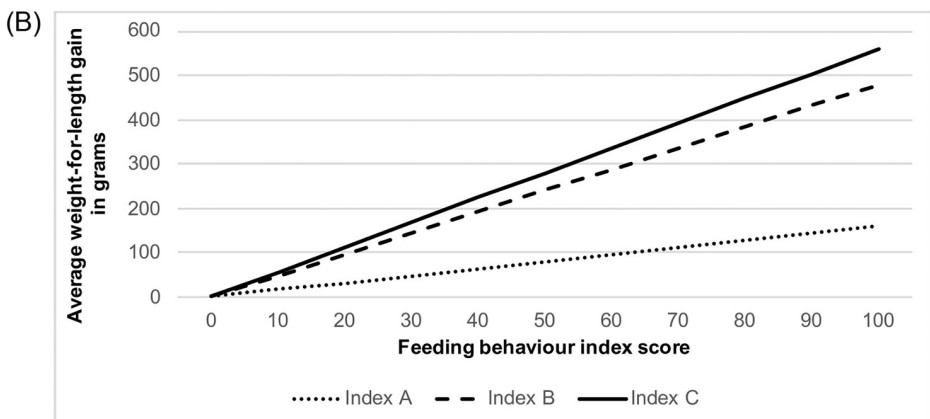

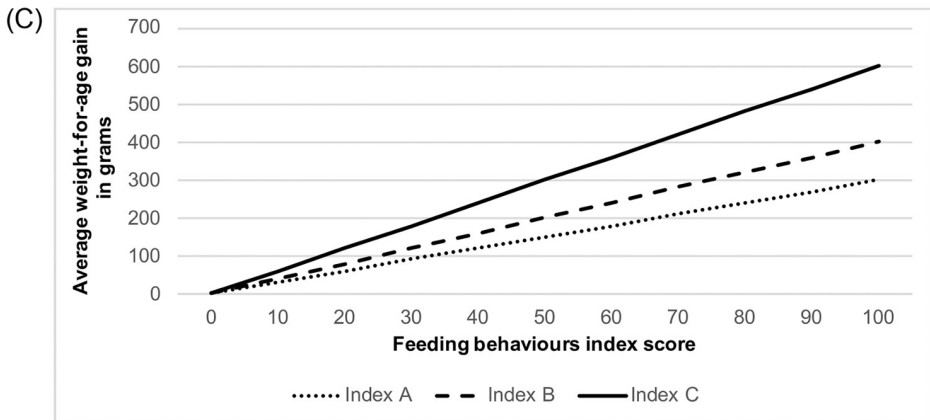

**Fig 4.** **(A)** Average length-for-age gain with increase in feeding behaviours score. **(B)** Average weight-for-length gain with increase in feeding behaviours score. **(C)** Average weight-for-age gain with increase in feeding behaviours score.

for assessment of feeding by trained non-specialists in community-based interventions in low-resource settings.

On the whole, children showed little interest in food during mealtimes. This has been noted previously in similar settings [20,32] and may be connected to poor appetite. Those who

showed more responsive feeding behaviours towards feeding had greater weight-for-age and length-for-age, which may reflect longer-term improvements in nutrition [22].

Caregivers had some behaviours suggestive of a "laissez-faire" feeding style; on the whole they did not promote self-feeding nor stop children from doing it. They did not appear to follow children's feeding cues and did not show many strategies to overcome food refusal. "Controlling" feeding style behaviours were rare, as suggested by the low prevalence of harshness and force-feeding behaviours.

Caregiver behaviours scores were associated with weight-for-length, which reflects child current nutritional status and is prone to short-term variation [22]. Most children did not self-feed, despite their psychomotor ability to do so from the age of 9 months [15]. One explanation may be that children need a long time to self-feed at that age, whereas caregivers have competing demands on their time due to day-to-day chores or work [35]. However, we observed a high prevalence of some responsive behaviours as defined in the WHO complementary feeding guidelines, such as interacting socially with children, minimizing distraction and encouraging the child to eat [16]. As responsive feeding has been linked to higher food acceptance [32], our results suggests that higher caregiver behaviours scores lead to weight gain in the short-term through increasing dietary intake.

Caregiver-child interaction score was higher than that of child & caregiver behaviours taken independently and was associated with weight-for-length and weight-for-age. Our results are in line with previous findings which showed that caregivers may compensate for a child's lack of interest in feeding by increasing their responsiveness [20]. In the short-term, compensation behaviours may promote rapid weight-for-length gain in children that would reflect on their weight-for-age. However, in the long-term, compensation behaviours may result in stressful experience for both caregivers and children [36], which may explain why we did not find an association with length-for-age.

Our study had several strengths. We used a whole population, representative sample, in rural India, an understudied population. We made attempts to limit the risk of Hawthorne effect and expect meal observed to reflect usual caregiver and child behaviours during feeding.

Limitations are inherent to the study design and data availability. We observed only one meal per infant. Although we found that most meals were typical in terms of food provided, feeder and place of feeding, for some children, the meal observed was different from that of their usual feeding environment. The relationship of feeding styles to anthropometrical measures may be bidirectional, and this could not be assessed in this cross-sectional study. Despite considerable attempts to consider confounding, residual confounding cannot be ruled out; specifically, data were not available on birthweight and recent infection.

Our results show that feeding behaviours of children and caregivers, as well as caregiver-child interaction during feeding is associated with early childhood anthropometrical measures at only 12 months of age. There is an urgent need to support optimal child growth at this crucial age, and our new tool, alongside our initial findings provide support and a potential method of evaluation for further work in LMICs towards ensuring that all children have the opportunity to thrive.

## Supporting information

**S1 Appendix. Observed feeding tool.**
(DOCX)

**S2 Appendix. Observed feeding—Standard operating procedures.**
(DOCX)

## Acknowledgments

We gratefully acknowledge: Sangath, the implementing organisation in India, Wellcome Trust Bloomsbury Centre for Global Health Research policy group, David Mabey & Tamara Hurst (fellowship support to SB); Angela Vega & Despoina Xenikaki (Administrative support). We thank the SPRING Team in the UK & Pakistan who were generous with their time and advice (Sarmad Aziz, Neha Batura, Assad Hafeez, Zelee Hill, Raghu Lingam, Atif Rhaman, Shamsa Rizwan, Siham Sikander, Jolene Skordis-Worrall), and the following members of the SPRING TSC & DSMB (Rajiv Bahl, Jose Martines, Linda Richter, Therese Stukel, Susan Walker). Finally, we acknowledge the outcome assessors who learnt and used this tool in Rewari in the SPRING trial, and the many families who joined in with SPRING activities over a period of several years.

## Author Contributions

**Conceptualization:** Pauline Boucheron, Sunil Bhopal, Betty Kirkwood.

**Data curation:** Pauline Boucheron, Sunil Bhopal, Reetabrata Roy.

**Formal analysis:** Pauline Boucheron, Sunil Bhopal.

**Funding acquisition:** Sunil Bhopal, Betty Kirkwood.

**Investigation:** Pauline Boucheron, Sunil Bhopal, Deepali Verma, Reetabrata Roy, Divya Kumar, Gauri Divan, Betty Kirkwood.

**Methodology:** Pauline Boucheron, Sunil Bhopal, Deepali Verma, Reetabrata Roy, Divya Kumar, Gauri Divan, Betty Kirkwood.

**Supervision:** Sunil Bhopal, Gauri Divan, Betty Kirkwood.

**Writing – original draft:** Pauline Boucheron, Sunil Bhopal.

**Writing – review & editing:** Pauline Boucheron, Sunil Bhopal, Deepali Verma, Reetabrata Roy, Divya Kumar, Gauri Divan, Betty Kirkwood.

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
