## [Decision Letter · Decision Letter 0]

27 May 2020

PONE-D-20-05725

Observed feeding behaviours and effects on child growth at 12 months of age: findings from the SPRING cluster-randomized controlled trial in rural India

PLOS ONE

Dear Dr. Boucheron,

Thank you for submitting your manuscript to PLOS ONE. We believe the manuscript has merit but does not fully meet PLOS ONE’s publication criteria as it currently stands. Therefore, we invite you to submit a revised version of the manuscript that addresses the points raised during the review process. The comments of the two reviewers is attached below. Please address especially the question of review 1 on trial arm allocation and feeding behaviour. 

We look forward to receiving your revised manuscript.

Kind regards,

Frank Wieringa, M.D., Ph.D.

Academic Editor

PLOS ONE

Journal Requirements:

Reviewers' comments:

Reviewer's Responses to Questions

**Comments to the Author**

1. Is the manuscript technically sound, and do the data support the conclusions?

Reviewer #1: Yes

Reviewer #2: Yes

2. Has the statistical analysis been performed appropriately and rigorously? 

Reviewer #1: Yes

Reviewer #2: Yes

3. Have the authors made all data underlying the findings in their manuscript fully available?

Reviewer #1: No

Reviewer #2: Yes

4. Is the manuscript presented in an intelligible fashion and written in standard English?

Reviewer #1: Yes

Reviewer #2: Yes

5. Review Comments to the Author

Reviewer #1: General comment

This draft article reports a cross-sectional observational study of 857 12-month-old children and their mother. This study is nested in the SPRING RCT in North India since 2013. The objective is to assess a new tool to describe child and caregiver behaviours with regards to responsive feeding recommendations and define corresponding scores, and then look after potential associations between these scores and anthropometric z-scores. Data are scarce on this subject and this article reports an important and interesting work.

However, the presentation of the article could be improved as in its current state, it is presented more like a report with stress put on the methods than like a scientific article. Some parts could be shortened, while other could be detailed further.

One question is why the association between trial arm allocation and feeding behaviours was not assessed in this study. When modelling the association between feeding behaviours and anthropometric outcomes, it is reported that data were adjusted in accordance (intervention vs control) but nothing is said about the size and the significance of this effect.

Thereafter are some more detailed comments given section by section.

Title: should be changed, as the study does not report any follow-up of growth. The text of the article is much more cautious about this aspect, reporting only associations between mother and child behaviours during meal and anthropometric status.

Abstract

Line 67: The statistical method used to look for association between behaviours and anthropometric data should appears in the paragraph “methods”.

Line 69 : age of children should be indicated here (this first sentence would rather be in the method paragraph)

Line 81: it is not clear what authors mean by “different elements of growth”?

Introduction:

Lines 114-120: this section should be developed a bit more. What authors are meaning by standard methods? Some articles have reported standardized methods to describe and assess mother and child behaviours during meals. They should be referenced here (some are referenced subsequently, and some are missing.) What is really new is the method used for scoring.

Line 130: not impact but rather association

Material & method section:

Line 137: title should rather be: Overview of the SPRING study design

Lines 140-141: a few words about the content of the SPRING intervention would be appreciated here or in the introduction section.

Line 165: a reference is missing

Line 169: the sampling method specifically used for this study (in the § sample size) should be described here rather than after methods of data collection

Line 178: was the assessment precisely done on the day of children’s first birthday? Or was there a range for assessment?

Lines 180-192: description of methods for anthropometric assessment could be shortened here and a reference to the WHO’ s standardized methods should be added.

Line 194: how long did the assessors stay in the home?

Line 196/Table 1: does the study presented here use the results of all 34 items? To avoid confusion, the whole table (or questionnaire) could be provided as supplemental data, and table 1 could focus only on the items used in this study with more details about the way of scoring of each item Or maybe table 2 is this focused table? For example: the way of scoring self-feeding C1 and C2 is not clear. For C2, score should be negative ? (same for C5, C6). Is the choice of 0, 1, 2 or 3 related to the frequency of the presence of a given behaviour? This is the responsibility of the assessor?

Lines 204-221: this paragraph could be shortened (for example lines 204-205 are useless). The page number should be removed.

Lines 205-206 and 211-212 (and E6 in table 1): mouthful were counted but no data are provided in the results ‘section about this. This is a pity because the effect of feeding behaviours could be mediated by an effect of behaviours about food intakes. Data on the number of katoris are neither given. Please indicate the volume of a katori.

Table 1, E9: Replace inside by outside the courtyard

Lines 223-226: should be removed if no results about these questionnaires are given.

Lines 243-245: ref. 32 cited twice in the same sentence.

Table 2: Invert the caregivers behaviours (index B) and the child behaviours (index A), as it is confusing. The way of scoring each item should be explained here, and the maximum score for each category (or indice).

Lines 260-261: each behaviour was converted in a binary variable. Does this mean that all behaviour had the same weight in the score? Promoting means +1 and discouraging – 1?

Lines 278-279: the only words about the effect of trial arm allocation is here! Even if this is not the main purpose of the study, knowing if there was an effect of intervention on feeding behaviours, could tell us about the possibility of changing these behaviours.

Lines 283-284: Indeed, increased food intakes is one of the awaited effect of the responsive feeding which could have an effect on nutritional status, so having a look on the effect of feeding behaviours on food intakes would be interesting.

Results section

Figure 1. To make the reading easier, the flowchart could be completed to go until the included observation

Table 3: which arm is the one of intervention? Of control?

Line 331: figures 2 A, B, C are not only histograms. Only the figure 2A is related to length-for age.

The three figures could be deleted as information in table 4 is sufficient, or if authors want to keep them, they could be reunited in a single figure, of smaller size

Lines 332-333: could be deleted

Line 336: …had signs of moderate to severe undernutrition

Line 338: low length-for-age z-score, low weight-for-age z-score, low weight-for-length z-score

Figures 2A, B C: there is no red line. Also add z-score where needed.

Lines 371-372: we would like to know which components. The 3 figures of PCA could be displayed to help the reader. Does this refer to table 2? If so, we can count only 5 comments in table 2 b for child behaviours (but indeed 11 for index B)

Line 378: 75 % + 57% are not 100%, please explain

Table 5 is useless as median scores of feeding behaviour indices are already in the text (line 368)

Line 396: …Please change in …with a positive linear change in length-for-age Z-score of 0.004

Table 6: the three stars given as table footnote are not in the table.

Figures 4A, B, C: Figure captions and titles of ordinate axes should be changed: the unit for anthropometric z-scores is the number of SD, and cannot be cm or g. Since all observations were done at the same age, a correspondence can be established between length-for-age z-score change with length change (in cm), or between weight-for-age z-score change with weight change (in g). But this is not possible for weight-for-length.

Discussion:

The repetition of results presented in the results’ section should be avoided here, and the results could be more deeply discussed with regards of the literature on this subject.

Lines 446-447: “those who showed more interest…improvements in nutrition” On which analysis is based this statement?

Lines 449-451 vs 458-459: these statements seem contradictory: behaviours suggestive as laissez-faire style /high prevalence of responsive behaviours?

Line 465: does a higher score means a greater utilization here?

Line 474: as almost half of the initial sample was not included for different reasons, are the authors sure that the study sample is still representative?

Only one meal per infant was observed, could this be a limitation?

References:

Check if the references are presented in a way accepted by Plos One. The current format is not easy to read. The reference 11 also should be checked.

Reviewer #2: This was a well-described study exploring the association between feeding practices and child anthropometric outcomes. I have a number of comments relating to the statistical analysis and reporting.

Abstract: where you report mean anthropometric measures please specify the quantities indicated in brackets; are they SDs or SEs? They should ideally be SEs since anthropometric measures are your outcomes. It is also unclear in the results what 'caregiver behaviours' and 'child behaviours' means - this seems to me to be a qualitative factor, therefore I am struggling to understand, on the basis of the abstract alone, how it can relate to quantitative outcomes. You need to be clear what the quantitative aspect of behaviour is being correlated with anthropometric characteristics. Having read the rest of the manuscript, I suspect you may mean 'more responsive feeding behaviour was associated with increased WFL etc'; if this is the case, you should report as such in the abstract, and also report the beta-coefficients as differences in outcomes per unit increase in responsive behaviour score.

Methods:

Table 1 please indicate the directionality of the quantitative scores; for example where items were scored on a scale of 0 to 30 or 0 to 3, what do higher/lower scores indicate?

Exposures: please describe how the thresholds for converting item scores into binary responses were determined.

Please describe whether your PCA was based on the default Pearson correlation coefficients or whether you used tetrachoric correlation coefficients; the latter are the appropriate choice for PCA of binary items.

Results:

A descriptive table summarising the characteristics of participants who were observed in both arms, with counts (%) for categorical variables and means (SDs) for continuous ones, should be included and briefly summarised in the first or second paragraph of the sample description, after talking about the flow diagram.

Table 3: means of continuous outcomes should be accompanied by SEs (more appropriate for inference, which is what is being done here) rather than SDs (which are descriptive).

Table 4: as this table is reporting outcomes, the means should be accompanied with SEs not SDs.

Line 364 - I suppose you mean 'the first principal component of index...' in each case; I would expect that you extracted the first PCs in each case.

The counts and proportions for the binary items which are described in the paragraph starting at line 360 should also be tabulated.

The beta-coefficients in table 6 are differences and should be referred to as such here and elsewhere in the manuscript.

6. PLOS authors have the option to publish the peer review history of their article (what does this mean?). If published, this will include your full peer review and any attached files.

Reviewer #1: No

Reviewer #2: No

---

## [Author Response · Author response to Decision Letter 0]

2 Jul 2020

Review Comments to the Author

Reviewer #1: General comment

This draft article reports a cross-sectional observational study of 857 12-month-old children and their mother. This study is nested in the SPRING RCT in North India since 2013. The objective is to assess a new tool to describe child and caregiver behaviours with regards to responsive feeding recommendations and define corresponding scores, and then look after potential associations between these scores and anthropometric z-scores. Data are scarce on this subject and this article reports an important and interesting work.

However, the presentation of the article could be improved as in its current state, it is presented more like a report with stress put on the methods than like a scientific article. Some parts could be shortened, while other could be detailed further.

We thank the reviewer for these positive comments regarding our work. We have edited in line with suggestions in the detailed comments below.

One question is why the association between trial arm allocation and feeding behaviours was not assessed in this study. When modelling the association between feeding behaviours and anthropometric outcomes, it is reported that data were adjusted in accordance (intervention vs control) but nothing is said about the size and the significance of this effect.

Whilst this is not the focus of this paper and will be tackled more comprehensively alongside other measures of feeding knowledge, understanding and behaviours, we have now analysed the impact of trial arm allocation on feeding behaviours as requested. Results are as follows:

- Impact of SPRING trial arm on overall feeding behaviour score, adjusted for all the other potential confounders (fully adjusted): 0.02 (95%CI: -0.18;0.23), p=0.83

- Impact of SPRING trial arm on child feeding behaviour score, adjusted for all the other potential confounders (fully adjusted): 0.02 (95%CI: -0.18;0.23), p=0.82

- Impact of SPRING trial arm on caregiver feeding behaviour score, adjusted for all the other potential confounders (fully adjusted): 0.02 (95%CI: -0.18;0.22), p=0.85

We have added the following comment to the manuscript on new lines 359-361 in the cleaned version of the revised manuscript: “SPRING trial arm allocation did not have a meaningful impact on feeding behaviours, with very small point estimates, wide confidence intervals and large p-values for each of the feeding scales (data not shown).” 

Thereafter are some more detailed comments given section by section.

Title: should be changed, as the study does not report any follow-up of growth. The text of the article is much more cautious about this aspect, reporting only associations between mother and child behaviours during meal and anthropometric status.

We have replaced ‘growth’ with ‘weight and length’.

Abstract

Line 67: The statistical method used to look for association between behaviours and anthropometric data should appears in the paragraph “methods”.

We have added this on new lines 68-69.

Line 69 : age of children should be indicated here (this first sentence would rather be in the method paragraph)

We have clarified this in new line 62 and 65.

Line 81: it is not clear what authors mean by “different elements of growth”?

We have replaced this phrase with ‘weight and length’ on new lines 83-84.

Introduction:

Lines 114-120: this section should be developed a bit more. What authors are meaning by standard methods? Some articles have reported standardized methods to describe and assess mother and child behaviours during meals. They should be referenced here (some are referenced subsequently, and some are missing.) What is really new is the method used for scoring.

We have clarified this on new lines 117-118: “This is because there is no method which allows non-specialists to perform a holistic responsive feeding assessment”.

We have added reference to four previous methods for doing elements of a feeding assessment. 

We have emphasized where our method adds value in new lines 132-135: “we first present a method that uses a new scoring system to develop three indices for measurement of 1) child behaviours 2) caregiver behaviours and 3) caregiver-child interaction during feeding from a larger tool.”

Line 130: not impact but rather association

We agree and have edited this on new line 135.

Material & method section:

Line 137: title should rather be: Overview of the SPRING study design

We agree and have edited on new line 142.

Lines 140-141: a few words about the content of the SPRING intervention would be appreciated here or in the introduction section.

Thank you. This is now added in lines 146-147.

Line 165: a reference is missing

Thank you. This is now added on new line 167.

Line 169: the sampling method specifically used for this study (in the § sample size) should be described here rather than after methods of data collection

We have replaced the eligibility criteria with the sample size calculation and sampling method to aid clarity for the reader on new lines 173 -178.

Line 178: was the assessment precisely done on the day of children’s first birthday? Or was there a range for assessment?

We have added within “-7 to +21 days of this birthday” to aid clarity (new line 188).

Lines 180-192: description of methods for anthropometric assessment could be shortened here and a reference to the WHO’ s standardized methods should be added.

Thank you, we have done so on new line 197.

Line 194: how long did the assessors stay in the home?

We have added this on new line 200.

Line 196/Table 1: does the study presented here use the results of all 34 items? To avoid confusion, the whole table (or questionnaire) could be provided as supplemental data, and table 1 could focus only on the items used in this study with more details about the way of scoring of each item Or maybe table 2 is this focused table? 

Thank you for your comment. The study focuses on Caregiver’s behaviours, child’s behaviours and caregiver-child interaction observed during the meal. The whole formatted questionnaire with details on scoring for each item is presented in supplemental data 1 (S1). To avoid confusion, we suppressed Table 1, stated that the whole tool was presented in S1 with details on scoring per each item, and kept Table 2 (new Table 1) because this table focuses on items used in this paper. 

For example: the way of scoring self-feeding C1 and C2 is not clear. For C2, score should be negative? (same for C5, C6). Is the choice of 0, 1, 2 or 3 related to the frequency of the presence of a given behaviour? This is the responsibility of the assessor?

We have added the following to new Table 1 caption to clarify this: “Table 1. Items* of the Observed feeding Tool included in feeding behaviours indices**,***.

*Each item included in the indices were made binary, scored 1 if the behaviour was observed and 0 if it was not observed

**Each index was scored on a scale from 0 (lowest score corresponding to the least responsive feeding behaviours) to 100 (highest score corresponding to the highest responsiveness during feeding)

***All child and caregiver items were included in the Caregiver-child interaction index. ”

We needed to reverse child feeding behaviours score. This was not automatically done by the PCA. The direction of child score was opposite to that of caregiver or overall scores.

We have added the following to the manuscript on new lines 254-255: “The child feeding behaviour scores were reversed because positive scores here indicated poorly quality feeding behaviours.”

Lines 204-221: this paragraph could be shortened (for example lines 204-205 are useless). The page number should be removed.

Thank you for your comment. We removed lines 204-205 and referred to Appendices S1 (the whole Observed Feeding Tool) and S2 (Standard Operating Procedures for the meal observation) (new lines 201-202), and removed page numbering, which allowed us to shorten the paragraph. 

Lines 205-206 and 211-212 (and E6 in table 1): mouthful were counted but no data are provided in the results ‘section about this. This is a pity because the effect of feeding behaviours could be mediated by an effect of behaviours about food intakes. Data on the number of katoris are neither given. Please indicate the volume of a katori.

We indicated the volume of a katori on new line 217: “assessors estimated the volume of food consumed during the meal using a katori – a widely used small stainless-steel bowl with a volume of 160mL”.

We have added data on feeding quantity to the results section under the paragraph “Sample Description”:

- New Table 2: Description of number of mouthful (median, IQR) and number of katoris (%, n) of food eaten by children observed during a meal

- New lines 312-314: “Children ate a median of 13 mouthfuls (IQR=9;19) of food and most of them ate less than a 1/4 of a standard katori (n=431/857, 50.3%), representing a volume of food of about 40mL”.

However, as we wanted to study the effect of mealtime behaviours on anthropometric outcomes, “we did not consider food quantity or food diversity as confounders because these were likely to be on the causal pathway [35,36].” (This is stated in lines 273-274).

Table 1, E9: Replace inside by outside the courtyard

Both of these refer to being inside a courtyard with the difference being only the flooring type (paved vs mud/dust). 

Lines 223-226: should be removed if no results about these questionnaires are given.

Thank you for your comments. We have deleted these lines. 

Lines 243-245: ref. 32 cited twice in the same sentence.

Deleted first instance.

Table 2: Invert the caregivers behaviours (index B) and the child behaviours (index A), as it is confusing. The way of scoring each item should be explained here, and the maximum score for each category (or indice).

Thank you for your comment. We inverted Index A (child behaviours) and B (caregiver behaviours) in new Table 1. 

We describe the way of scoring items below: 

- First step: Each item listed in Table 1 was scored by the assessor during mealtime, as described in Appendix S1. 

- Second step: Items were transformed into binary for PCA (0 if the behaviour was not observed, 1 if it was observed by the observer during the mealtime)

- Third step: We ran the PCA for each index and the raw score obtained was standardized on a scale from 0 (lowest possible score) to 100 (highest possible score). 

To clarify for the reader, we changed the title of new Table 1 to “Items* of the Observed feeding Tool included in feeding behaviours indices**,***”, a) Index A: Child behaviours, b) Index B: Caregiver behaviours”, and added notes under the table : 

- *Each item included in the indices were made binary, scored 1 if the behaviour was observed and 0 if it was not observed

- **Each index was scored on a scale from 0 (lowest score corresponding to the least responsive feeding behaviours) to 100 (highest score corresponding to the highest responsiveness during feeding)

- ***All child and caregiver items were included in the Caregiver-child interaction index

Lines 260-261: each behaviour was converted in a binary variable. Does this mean that all behaviour had the same weight in the score? Promoting means +1 and discouraging – 1?

As described above, a behaviour observed at least one during mealtime was coded +1, independently of it being a promoting or discouraging behaviour. A behaviour that was not observed during mealtime was coded 0.

Lines 278-279: the only words about the effect of trial arm allocation is here! Even if this is not the main purpose of the study, knowing if there was an effect of intervention on feeding behaviours, could tell us about the possibility of changing these behaviours.

It is important that the impact of the SPRING trial on these behaviours is not overemphasised as other work is in progress describing the trial and its impact in more detail across a range of outcomes. We do not think a detailed understanding of the trial impact is key to understanding this work. However, we have presented data on lines 359-361 as follows: “SPRING trial arm allocation did not have a meaningful impact on feeding behaviours, with very small point estimates, wide confidence intervals and large p-values for each of the feeding scales (data not shown).”

Lines 283-284: Indeed, increased food intakes is one of the awaited effect of the responsive feeding which could have an effect on nutritional status, so having a look on the effect of feeding behaviours on food intakes would be interesting.

You are right that there is an impact of feeding behaviours on food intakes:

- Average increase in number of mouthful intake per unit increase in Child feeding behaviour score: 0.40 (95%CI:0.21;0.59), p<0.0001 

- Average increase in number of mouthful intake per unit increase in Caregiver feeding behaviour score: 0.36(95%CI :0.24 ;0.48), p<0.0001 

- Average increase in number of mouthful intake per unit increase in Overall feeding behaviour score: 0.54 (95%CI: 0.42;0.66), p<0.0001 

However, as we wanted to study the effect of mealtime behaviours on anthropometric outcomes, and therefore “we did not consider food quantity or food diversity as confounders because these were likely to be on the causal pathway [35,36].” (This is stated in lines 273-274). Taking into account food intakes would therefore have biased the estimates of the impact of feeding behaviours on anthropometric outcomes towards the null. In consequence, we think this is out of scope of this paper and did not report these data in the paper. 

Results section

Figure 1. To make the reading easier, the flowchart could be completed to go until the included observation

We like the flowchart and think it aids the reader in understanding the representativeness of the sample. Could the reviewer rephrase their concern if it still stands.

Table 3: which arm is the one of intervention? Of control?

The authors are blind to trial arm allocation and this table is simply to check for evidence of selection bias.

Line 331: figures 2 A, B, C are not only histograms. Only the figure 2A is related to length-for age.

We have removed the reference to histograms and made it clear that the figure reference refers to the whole paragraph (new lines 335-336). 

The three figures could be deleted as information in table 4 is sufficient, or if authors want to keep them, they could be reunited in a single figure, of smaller size

Thank you for your comment. We combined the three figures into one of a smaller size (new Fig. 2).

Lines 332-333: could be deleted

We deleted these lines. 

Line 336: …had signs of moderate to severe undernutrition

We added “had signs of moderate to severe undernutrition” to line 330.

Line 338: low length-for-age z-score, low weight-for-age z-score, low weight-for-length z-score

We added “low length-for-age z-score, low weight-for-age z-score, low weight-for-length z-score” to new lines 331-333.

Figures 2A, B C: there is no red line. Also add z-score where needed.

Thank you for your comment. We corrected and reedited the figure. 

Lines 371-372: we would like to know which components. The 3 figures of PCA could be displayed to help the reader. Does this refer to table 2? If so, we can count only 5 comments in table 2 b for child behaviours (but indeed 11 for index B)

Each behaviour in new Table 1 corresponds to each component of the PCA. Child feeding behaviours are composed of 6 behaviours, and there are 11 for caregiver feeding behaviours score. 

Line 378: 75 % + 57% are not 100%, please explain

These numbers represent the proportion of children who were seen by observers as doing at least once two different types of feeding behaviours (as described in new Table 1):

- 75% of children showed disinterest in food at least once during mealtime, by saying no, sticking out their tongue, closing their mouth, turning or moving away 

- 57% of children tried to get food, meaning that they showed interest in food during mealtime, by asking for food, pointing to food, reaching for food, touching food or opening their mouth

As these two behaviours are not strictly opposites, and to avoid confusion, we changed the term “showing interest in food” by “trying to get food”. 

Table 5 is useless as median scores of feeding behaviour indices are already in the text (line 368)

We have removed Table 5. 

Line 396: …Please change in …with a positive linear change in length-for-age Z-score of 0.004

Thank you for your comment. We changed to “with a positive linear difference in length-for-age Z-score of 0.004” on line 376, and similarly on lines 383 and 391. We used the term difference instead of change to answer another reviewer’s comment.

Table 6: the three stars given as table footnote are not in the table.

We have now corrected this in Table 6 (new Table 5) 

Figures 4A, B, C: Figure captions and titles of ordinate axes should be changed: the unit for anthropometric z-scores is the number of SD, and cannot be cm or g. Since all observations were done at the same age, a correspondence can be established between length-for-age z-score change with length change (in cm), or between weight-for-age z-score change with weight change (in g). But this is not possible for weight-for-length.

The purpose of these graphs is to present the already-presented data in more meaningful manner to the generalist reader. We would prefer to leave them as they are currently, however are open to removing them if this is deemed necessary by the reviewers/editor.

You are right that it is possible to obtain a correspondence between length-for-age Z-score change with length change, or between weight-for-age Z-score with weight change. We built our graphs in cm and grams based on the data from the WHO growth curves for both boys and girls at 12 months. 

Similarly, for weight-for-length, we were able to obtain a value in grams by taking as a starting point, the weight that children are expected to have for a mean height at 12 months, based on the WHO growth curves. 

The results we display are overall weighted means in length and weight increase for children at 12 months, for each anthropometric outcome. 

 Discussion:

The repetition of results presented in the results’ section should be avoided here, and the results could be more deeply discussed with regards of the literature on this subject.

Lines 438-440: “those who showed more interest…improvements in nutrition” On which analysis is based this statement?

There was a positive linear association between child responsive feeding behaviours score and length-for-age & weight-for-age, which are two markers of long-term nutritional status in children. We clarified this statement and changed to: “Those who showed more responsive feeding behaviours towards feeding had greater weight-for-age and length-for-age, which may reflect longer-term improvements in nutrition [22].” (new lines 423-425)

Lines 449-451 vs 458-459: these statements seem contradictory: behaviours suggestive as laissez-faire style /high prevalence of responsive behaviours?

There were both a high prevalence of some of the behaviours observed in caregiver that were suggestive of a laissez-faire feeding style (such as not promoting self-feeding nor stopping the child from doing it), while other behaviours were suggestive of a responsive feeding style (such as interacting socially with children, minimizing distraction and encouraging the child to eat). We clarified these statements and changed to: “Caregivers had some behaviours suggestive of a “laissez-faire” feeding style; on the whole they did not promote self-feeding nor stop children from doing it.” on new lines 426-427; and “However, we observed a high prevalence of some responsive behaviours as defined in the WHO complementary feeding guidelines, such as interacting socially with children, minimizing distraction and encouraging the child to eat” on new lines 435-438. 

Line 465: does a higher score means a greater utilization here?

All child and caregiver items have been put into the PCA to construct the caregiver-child interaction score. We therefore expect this score to reflect interaction between caregiver and child. Therefore, a higher score is expected to be the result of more positive interactions between child and caregiver. It can reflect how child adapts to the caregiver and how caregiver adapts to the child during feeding taking into account how “the other” behaves (caregiver for the child, and child for the caregiver).

Line 474: as almost half of the initial sample was not included for different reasons, are the authors sure that the study sample is still representative?

There was no evidence of difference between those included and not included by trial arm, maternal education, SES, child sex, delivery in a facility, and maternal age at delivery.

It is rare to have the opportunity to do this sort of population-based sampling and we think the reader will decide how it compares to other more purposive methods of sampling.

Only one meal per infant was observed, could this be a limitation?

Yes, we have added this on lines 455-458: “We observed only one meal per infant. Although we found that most meals were typical in terms of food provided, feeder and place of feeding, for some children, the meal observed was different from that of their usual feeding environment.”

References:

Check if the references are presented in a way accepted by Plos One. The current format is not easy to read. The reference 11 also should be checked.

We corrected the reference 11.

We have checked the referencing guide from PLOS ONE available at https://journals.plos.org/plosone/s/submission-guidelines#loc-references. The references appear to us to follow this guidance. We seek further guidance from journal staff.

Reviewer #2: This was a well-described study exploring the association between feeding practices and child anthropometric outcomes. I have a number of comments relating to the statistical analysis and reporting.

We thank the reviewer for their positive comments and provide details responses below.

Abstract: where you report mean anthropometric measures please specify the quantities indicated in brackets; are they SDs or SEs? They should ideally be SEs since anthropometric measures are your outcomes.

We have now specified in the abstract and in the main text that the quantities in brackets are SDs which we believe are an appropriate method for showing the ‘spread’ of these data around their mean.

It is also unclear in the results what 'caregiver behaviours' and 'child behaviours' means - this seems to me to be a qualitative factor, therefore I am struggling to understand, on the basis of the abstract alone, how it can relate to quantitative outcomes.

You need to be clear what the quantitative aspect of behaviour is being correlated with anthropometric characteristics. Having read the rest of the manuscript, I suspect you may mean 'more responsive feeding behaviour was associated with increased WFL etc'; if this is the case, you should report as such in the abstract, and also report the beta-coefficients as differences in outcomes per unit increase in responsive behaviour score.

We have now done this.

Methods:

Table 1 please indicate the directionality of the quantitative scores; for example where items were scored on a scale of 0 to 30 or 0 to 3, what do higher/lower scores indicate?

Exposures: please describe how the thresholds for converting item scores into binary responses were determined.

Thank you for your comment. We describe the way of scoring items below: 

- First step: Each item listed in Table 1 was scored by the assessor during mealtime, as described in Appendix S1. 

- Second step: Items were transformed into binary for PCA (0 if the behaviour was not observed, 1 if it was observed by the observer during the mealtime)

- Third step: We ran the PCA for each index and the raw score obtained was standardized on a scale from 0 (lowest possible score) to 100 (highest possible score). 

New Table 1 presents “Items of the Observed feeding Tool included in feeding behaviours indices, a) Index A: Child behaviours, b) Index B: Caregiver behaviours”: 

- Each item included in the indices were made binary, scored 1 if the behaviour was observed and 0 if it was not observed

- Each index was scored on a scale from 0 (lowest score corresponding to the least responsive feeding behaviours) to 100 (highest score corresponding to the highest responsiveness during feeding)

- All child and caregiver items were included in the Caregiver-child interaction index

Please describe whether your PCA was based on the default Pearson correlation coefficients or whether you used tetrachoric correlation coefficients; the latter are the appropriate choice for PCA of binary items.

We ran the PCA based on Pearson correlation coefficients in order to retain the same analysis principle throughout the SPRING trial including for socio-economic status, which was based on Vyas & Kumarayake (https://pubmed.ncbi.nlm.nih.gov/17030551/)’s work on construction of SES scores in LMICs.

Results:

A descriptive table summarising the characteristics of participants who were observed in both arms, with counts (%) for categorical variables and means (SDs) for continuous ones, should be included and briefly summarised in the first or second paragraph of the sample description, after talking about the flow diagram.

This is described in Table 3 (new Table 2). We added a paragraph to briefly summarise these results under the section “Sample Description” in new lines 307-311: “Table 2 presents sociodemographic characteristics of the 857 mother-child pairs observed during feeding. Overall, 442 children (51.6%) were males, and 20 (2.3%) were twins or triplets. Most children (n=838, 97.8%) had been delivered in facilities, of mothers with a mean age at delivery of 22.3 (SD=3.6) years. The majority of mothers had an education level of 10th to 12th grade (n=335, 39.1%)”. 

Table 3: means of continuous outcomes should be accompanied by SEs (more appropriate for inference, which is what is being done here) rather than SDs (which are descriptive).

The purpose of the standard deviations presented in new Table 2 is to help describe characteristics of mother-child pairs included in the study sample. The SDs are used here simply to describe the variance around the means. We do not agree with the reviewer that these should be SEs. There is no inference.

Table 4: as this table is reporting outcomes, the means should be accompanied with SEs not SDs.

The purpose of Table 4 (new Table 3) is to describe anthropometric characteristics of children included in our study sample. The SDs used here are simply to describe the variance around the sample mean presented. We do not agree with the reviewer that an SE is more appropriate for this purpose.

Line 364 - I suppose you mean 'the first principal component of index...' in each case; I would expect that you extracted the first PCs in each case.

Yes, this is correct, we extracted the first principal component for each index. We have added this on new line 251.

The counts and proportions for the binary items which are described in the paragraph starting at line 360 should also be tabulated.

Thank you for your comment, we tabulated the binary items and displayed the results in a table. These appear in new Table 4.

The beta-coefficients in table 6 are differences and should be referred to as such here and elsewhere in the manuscript.

Thank you for your comment. We have now done this.

---

## [Editor Report · Decision Letter 1]

23 Jul 2020

Observed feeding behaviours and effects on child weight and length at 12 months of age: findings from the SPRING cluster-randomized controlled trial in rural India

PONE-D-20-05725R1

Dear Dr. Boucheron,

We’re pleased to inform you that your manuscript has been judged scientifically suitable for publication and will be formally accepted for publication once it meets all outstanding technical requirements.

Kind regards,

Frank Wieringa, M.D., Ph.D.

Academic Editor

PLOS ONE
---

## [Editor Report · Acceptance letter]

29 Jul 2020

PONE-D-20-05725R1 

Observed feeding behaviours and effects on child weight and length at 12 months of age: findings from the SPRING cluster-randomized controlled trial in rural India 

Dear Dr. Boucheron:

I'm pleased to inform you that your manuscript has been deemed suitable for publication in PLOS ONE. Congratulations! Your manuscript is now with our production department. 

Kind regards, 

on behalf of

Dr. Frank Wieringa 

Academic Editor

PLOS ONE